# Outcomes in Acute Decompensated Congestive Heart Failure Admissions with Chronic Liver Disease: A Nationwide Analysis Using the National Inpatient Sample

**DOI:** 10.3390/medsci13010019

**Published:** 2025-02-13

**Authors:** Vivek Joseph Varughese, Vignesh Krishnan Nagesh, Pratiksha Moliya, Nelson Gonzalez, Emelyn Martinez, Hata Mujadzic, Maggie James, Abraham Lo, Simcha Weissman

**Affiliations:** 1Department of Internal Medicine, University of South Carolina, Easley, SC 29640, USA; nelson.gonzalez@uscmed.sc.edu (N.G.); hata.mujadzic@prismahealth.org (H.M.); 2Department of Internal Medicine, Hackensack Palisades Medical Center, North Bergen, NJ 07047, USA; emelyn03.em@gmail.com (E.M.); mag.james95@gmail.com (M.J.); abraham.lo@gmail.com (A.L.); simchaweissman@gmail.com (S.W.); 3Department of Gastroenterology and Hepatology, University of Nebraska, Omaha, NE 68001, USA; pratiksha_moliya1992@yahoo.in

**Keywords:** heart failure, chronic liver disease, cirrhosis, mortality, NIS

## Abstract

AIM: The aim of our study was primarily to analyze hospital outcomes for acute decompensated heart failure (ADHF) admissions with a comorbid diagnosis of chronic liver disease (CLD). METHODS: The NIS was used to select ADHF admissions. The population characteristics of general ADHF admissions were compared with ADHF admissions with a comorbid diagnosis of CLD. Multivariate probit logistic regression was used to analyze the association between a documented diagnosis of CLD/alcoholic liver disease and all-cause mortality in ADHF admissions. Confounders were accounted for. Propensity scoring and nearest neighbor matching were conducted to select a matched cohort with and without CLD from ADHF admissions to further look at mortality outcomes. RESULTS: ADHF admissions with a comorbid diagnosis of CLD had a significantly higher proportion of all-cause mortality, 0.054 (0.053–0.057), a higher length of hospital stay, 6.95 days (6.84–7.06), and a higher mean of total hospital charges, USD 88,068.1, when compared to ADHF admissions without a comorbid diagnosis of CLD: all-cause mortality, 0.045 (0.044–0.046); length of hospital stay, 6.18 days (6.13–6.23); and mean total hospital charges, USD 79,946.21. A comorbid diagnosis of CLD had a significant association with all-cause mortality in ADHF admissions: OR 1.23 (1.17–1.29) after accounting for confounders. In the propensity-matched cohorts, the cohort with a diagnosis of CLD from the ADHF admissions had a higher proportion of all-cause mortality, 0.042 (0.036–0.049), when compared to the cohort without a diagnosis of chronic liver disease, 0.027 (0.022–0.033). CONCLUSIONS: In analyzing the mortality and healthcare utilization outcomes for ADHF admissions, the comorbid diagnosis of CLD is shown to have significantly higher all-cause mortality, higher length of hospital stay, and higher mean total charges when compared to ADHF admissions without a diagnosis of CLD. A documented diagnosis of CLD had a statistically significant association with all-cause mortality in ADHF admissions after accounting for confounding factors.

## 1. Introduction

Hospital admissions related to congestive heart failure (CHF) are one of the leading causes of healthcare resource utilization in the United States (US). Guideline-directed medical therapy (GDMT) is the cornerstone of pharmacological therapy for patients with heart failure with reduced ejection fraction (HFrEF) and consists of the four main drug classes: renin–angiotensin system inhibitors, evidence-based β-blockers, mineralocorticoid inhibitors, and sodium–glucose cotransporter 2 inhibitors [1]. Congestive heart failure (CHF) and CHF 30-day readmission rates have been a major focus of efforts to reduce healthcare costs in the recent era [2]. Looking at national averages, admissions for heart failure with preserved ejection fraction (HFpEF) are also a major contributor towards heart failure admissions. Volume overload in heart failure is an end-common pathway of a diverse number of diseases, including congenital heart disease, ischemic coronary artery disease, and arrhythmia, and factors like volume overload from renal dysfunction and hypertensive urgencies. Complex physiological factors determine the timing of systolic function reduction; however, all systolic heart failures are known to have a component of diastolic heart failure earlier in the course of pathophysiology. The result of congestion and/or poor perfusion results in negative effects on multiple organ systems. Patients with HF are often medically complex, with multiple comorbidities, a long list of medications, and a wide range of symptoms that can be attributed to several etiologies [2]. Advanced mechanical circulatory devices and monitoring systems have made the treatment of volume overload in CHF effective over the years; however, the healthcare resource utilization burden of these advanced devices as well as economic disparities still place GDMT as the cornerstone of volume management in CHF patients [3]. Recent studies have found that in acute decompensated heart failure (ADHF), right- and left-sided pressures generally start to increase before any notable weight changes take place preceding an admission. ADHF may be a problem of volume redistribution among different vascular compartments instead of, or in addition to, fluid shifts from the interstitial compartment. Thus, identifying the heterogeneity of volume overload would allow for the guidance of tailored therapy [4].

Likewise, chronic liver disease (CLD) is yet another leading cause of hospital admissions in the United States. Although they do not share much in common in terms of etiology, the symptom burden in both CHF and CLD are related to intravascular fluid status, as well as the redistribution of fluids within the body’s spaces. Associated malnutrition and generalized inflammatory states increase the symptom burden due to CLD, but the primary driver remains volume overload due to portal hypertension and reduced oncotic pressure in the intravascular compartment due to hypoalbuminemia [5]. Systemic complications of portal hypertension are varied and include esophageal and related varices, portal gastropathy, and splenomegaly. However, long-standing portal hypertension can lead to splanchnic vasodilation and related ascites [6]. This results in a generalized vasodilatory state, the overactivation of the renin–angiotensin–aldosterone (RAAS) system, and sodium and water retention due to the common mineralocorticoid pathway. This explains why mineralocorticoid inhibition becomes the mainstay of volume management in CLD [7].

RAAS inhibition in the management of CHF holds a role in remodeling prevention, thus decreasing the progression of the initial etiologic insult of raised intracardiac pressures. Also, mineralocorticoid receptor blockade becomes an option down the line for increased natriuresis [8]. However, in the case of CLD, the RAAS system aids in maintaining systemic blood pressures in the generalized vasodilatory/pro-inflammatory state. This makes the use of RAAS inhibition challenging in CHF patients with concomitant CLD. Hepatopulmonary syndrome, portopulmonary hypertension, hepatic hydrothorax, and cirrhotic cardiomyopathy are also potential interactions in CHF patients with CLD, further challenging volume management [9]. Irrespective of the differences in etiology, like mentioned before, the symptom burdens of both conditions are related to the common endpoint of sodium and water retention. Loop diuretics being employed in both conditions, and their use in patients with decompensated liver disease, hold a higher propensity for electrolyte abnormalities, further complicating volume management [9]. The generalized immunosuppression associated with liver failure increases the chances for sepsis, a major cause for hospital admissions and death for patients with CHF.

## 2. Methods

NIS was used to analyze the data. STAT 18 and SPSS 29 (with R 4.3) were used for the analysis. Admissions with the main diagnosis of ADHF were selected (*ICD10 codes I150*, *I50810*, *I50812*, *I50814*, *I5021*, *I5022*, *I5031*, *I5032*, *I502*, *I5084*, *I503*, *I5020*, *I5030*, *I509*, *I5023*, *I5033*, *I50813*, *I504*, *I97130*, *I5041*, *I5042*, *I5043*, *I13.2*, *I130*, *I1310*, and *I1311 were used to select the study population*). Subpopulation population analytics were performed. For the inpatient outcomes, probit logistic regression and multivariate logistic regression were used: all-cause mortality, total charges, and mean length of stay were the outcomes studied. Factors like smoking, obesity, hyperlipidemia, diabetes, and CKD, which are the factors with known adverse outcomes in CHF, were accounted for in the regression analysis to account for confounding. Regression outcomes for chronic liver disease and alcoholic liver disease were assessed, accounting for the above confounders. To reassess the outcomes of mortality due to the large difference in the sample, a propensity score match analysis was performed: a cohort of 3793 patients each, admitted for ADHF, matched exactly for age, obesity, smoking, hypertension, hyperlipidemia, and diabetes, was used to analyze the outcome difference in the presence of concomitant chronic liver disease. A two-tailed *p*-value < 0.05 was used to determine statistical significance.

## 3. Results

We identified 4,391,904 patients who were admitted with a diagnosis of acute decompensated congestive heart failure. In this population sample, 146,050 people had coexisting unspecified chronic liver disease and 59,260 of the admissions had a diagnosis of alcoholic cirrhosis. We identified 90 patients in the heart failure admission cohort who had a documented diagnosis of liver transplantation (Table 1).

In the subpopulation with coexisting ADHF and chronic liver disease, mean age was 64.364 (64.164–64.565): 66.7% were white, 15.97% were Black, and 11.72% were Hispanic. Analyzing the Charlson comorbidity, 54.0% of the population had a comorbidity index ≥ 5. While assessing for Charlson comorbidity index in the general CHF population, 32.4% had a Charlson comorbidity index score ≥ 5. Considering the median interquartile household income quartile for the admissions (using ZIP codes), 33.48% were in the first quartile, 25.99% in the second, 23.72% in the third, and 16.81% in the fourth. Analyzing the subpopulation of ADHF admissions with alcoholic cirrhosis of the liver, the mean age of the population was 61.9 (61.69–62.17): 65.6% were white, 15.35% were Black, and 13.7% were Hispanic. Tabulating the Charlson comorbidity index, 54.9% of the population had a Charlson comorbidity index score ≥ 5. Looking at the median household income quartile for the household (using ZIP codes), 35.53% were in the first quartile, 25.68% in the second, 22.93% in the third, and 15.87% in the fourth.

The next step of our study was to analyze the in-hospital outcomes in terms of the proportion of all-cause mortality, mean length of hospital stay, and mean of total charges for general ADHF admissions and comparing them with the outcomes in ADHF admissions with comorbid CLD. Further stratification and comparison of outcomes was performed in the ADHF admissions with specific comorbid diagnosis of alcoholic cirrhosis (Table 2).

Looking at the results from our analysis, ADHF admissions with a comorbid diagnosis of CLD had a significantly higher proportion of mortality, 0.054 (0.053–0.057), when compared to general ADHF admissions, 0.045 (0.044–0.046). The proportion of mortality was even higher in ADHF admissions with a comorbid diagnosis of alcoholic liver disease: 0.071 (0.067–0.076). The mean length of hospital stay for ADHF admissions with comorbid CLD was higher, 6.95 days (6.84–7.06), when compared to general ADHF admissions, 6.18 days (6.13–6.23). The mean length of hospital stay was significantly higher for ADHF admissions with alcoholic liver disease, 7.26 days (7.09–7.43), and this was statistically higher when compared to general ADHF admissions as well as ADHF admissions with a diagnosis of CLD. The mean total hospital charges for ADHF admissions with CLD/alcoholic liver disease was significantly higher, USD 88,068.1 (85,201.03–90,935.3), when compared to general ADHF admissions, USD 79,946.21 (77,644.87–82,247.54). This shows that the diagnosis of CLD/alcoholic liver disease worsened in-hospital outcomes, both in terms of survival and healthcare resource utilization when compared to general ADHF admissions.

To further analyze the crude comparison and avoid confounding factors, we performed a multivariate logistic regression accounting for age, sex, race, alcohol abuse, smoking, hyperlipidemia, obesity, and chronic kidney disease (CKD) to assess the association between all-cause mortality and diagnosis of CLD in ADHF admissions (Table 3).

Our analysis proved that diagnosis of CLD did have a significant association with all-cause mortality in ADHF admissions, with an OR of 1.23 (95% CI: 1.31–1.46), and the association remained significant after regressing for confounding factors like age, sex, race, alcohol use, smoking, hyperlipidemia, obesity, and CKD, with an OR of 1.23 (1.17–1.29).

To identify the strength of association between all-cause mortality and diagnosis of alcoholic liver disease in ADHF admissions, we performed multivariate logistic regression accounting for age, sex, race, alcohol abuse, smoking, hyperlipidemia, obesity, and chronic kidney disease (CKD) to assess the association between all-cause mortality and diagnosis of CLD in ADHF admissions (Table 4).

Our analysis proved that a comorbid diagnosis of alcoholic liver disease did have a significant association with all-cause mortality in ADHF admissions, with an OR of 1.82 (95% CI: 1.69–1.96), and the association remained significant after regressing for confounding factors like age, sex, race, alcohol use, smoking, hyperlipidemia, obesity, and CKD, with an OR of 1.63 (1.51–1.74).

Since the total number of ADHF admissions without coexisting CLD/alcoholic liver disease is much higher than the patients with CLD/alcohol-related liver disease, analysis outcomes, even with regression analysis, can be biased. To account for the bias, causal/association inferences were studied using propensity score and nearest neighbor matching (1:1 match with caliper measure 0.1) (SPSS 29 and R 4.3 was used for the analysis).

From the total ADHF admissions sample, two cohorts, with and without chronic liver disease, were selected, matched for smoking, obesity, alcohol abuse, age, race, hypertension, hyperlipidemia, and diabetes (exact matching was performed for age, race, smoking, alcohol abuse, and CKD). Probit multivariate logistic regression analysis was performed on the matched sample (overall balance test for matching: Hansen and Bowers, 2010; chi square: 14.579; *p*-value = 0.006) (relative multivariate imbalance Li before matching: 0.359; after matching: 0.259).

Looking at the results from the analysis of the propensity-matched cohorts (Table 5), the proportion of all-cause mortality in the CLD + cohort among the matched ADHF admissions was higher, with a statistical significance of 0.042 (0.036–0.049), when compared to the proportion of all-cause mortality in the CLD – cohort at 0.027 (0.022–0.033).

## 4. Discussion

Looking at the results of our analysis, chronic liver disease as well as alcoholic cirrhosis hold a statistically significant negative influence on heart failure admissions in terms of mortality, morbidity, and healthcare resource utilization. While looking through the literature, we came across published studies with similar results. Of note is a database analysis using the NIS 2016 by Khalid et al. [10]. Their propensity-matched and multivariate logistic regression analysis of 7595 patients with both ADHF and cirrhosis showed that mortality was almost two times higher in patients with ADHF and cirrhosis compared to patients without cirrhosis (*p* < 0.001). The mean hospital length of stay was 1.2 times higher in patients with ADHF and cirrhosis (*p*\0.001), but the mean total burden of hospitalization was not statistically significant. In another database analysis by Yazdanyar et al. [11] using the national readmission database 2012–2013, the adjusted odds ratio for 30-day mortality risk among heart failure admissions with documented cirrhosis was noted to be 1.3 (95% CI: 1.2–1.4). In our analysis, we looked at whether alcoholic cirrhosis holds a more potent adverse effect on heart failure outcomes when compared to general chronic liver diseases. This was based on the pathologic difference alcoholic cirrhosis can have when compared to general CLD [12] due to alcohol-induced dilated cardiomyopathy, an increased propensity for arrhythmias. and the general elevated mortality risks associated with alcohol abuse. In a literature review by Mantovani et al. [13], it was noted that adult individuals with NAFLD in the absence of severe obesity, hypertension, and diabetes have mildly increased left ventricular mass and early features of diastolic dysfunction. Coronary heart disease, cardiac hypertrophy, heart failure, atrial fibrillation, aortic valve sclerosis, mitral annulus calcification, QTc prolongation, etc., were other associated cardiac complications noted in NAFLD. In a real-world study by Nan Hee Kim et al. [14], 1886 patients with CT-proven cirrhosis and no known cardiovascular risk factors were selected from the Korean Genome Registry. It was noted that the presence of cirrhosis was associated with an elevated E/Ea ratio with decreased TDI Ea velocity in these patients with statistical significance. The pathologic effects of liver cirrhosis on cardiac musculature were analyzed further in a real-world study by VanWagner et al. [15], and it was noted that patients with NAFLD had lower early diastolic relaxation (e’) velocity (10.8 ± 2.6 vs. 11.9 ± 2.8 cm/s), higher LV filling pressures (E/e’ ratio: 7.7 ± 2.6 vs. 7.0 ± 2.3), and a lower global longitudinal strain pattern. The study also noted subclinical cardiac remodeling in these patients. All these prove an association between liver diseases and worsened systolic/diastolic function of the heart, beyond the clinical aspect of CLD being an important factor for medication non-adherence. This also provides an explanation at the molecular level for the findings in our study. Beyond interactions at the molecular level, the etiologic association of tachyarrhythmia-induced cardiomyopathies and liver diseases is yet another factor that could have a harmful impact in heart failure management. In a retrospective analysis by Ballestri et al. [16], it was noted that NAFLD itself, especially in its more severe forms, exacerbates systemic/hepatic insulin resistance, causing atherogenic dyslipidemia, and releases a variety of pro-inflammatory, pro-coagulant, and pro-fibrogenic medications that may play important roles in the pathophysiology of cardiac and arrhythmic complications.

While the above-mentioned studies give insights into the causal relationship between liver and cardiac dysfunction, which can play a role in the propensity for volume overload and exacerbation, based on our clinical practice, medication non-adherence remains the leading cause of heart failure exacerbations. Further analyzing the literature, we looked for published evidence on chronic liver diseases having a negative impact on the medication management of systolic heart failure. In a literature review by Robert W. Shrier [17], it was noted that third spacing/volume overload in both cardiac and liver failure is largely driven by the non-osmotic activation of the arginine vasopressin as well as the RAAS. Activation of the RAAS is driven by circulatory volume depletion in systolic heart failure, while it causes a decrease in systemic vascular resistance in liver dysfunction. The effect of renal sodium loss of aldosterone is seen in both conditions. However, natriuretic doses of aldosterone antagonists are used in liver failure, while non-natriuretic doses are employed in cardiac dysfunction. This could possibly be explained by the effects of hyperkalemia in cardiac dysfunction due to the concurrent use of ACE/ARBs. Although diuretics tend to be useful in different edematous conditions, the effect of diuretics in cirrhosis and cardiac failure is mainly mediated by pharmacodynamics, while in conditions like CKD, it is mediated by the pharmacokinetic effect [18,19]. However, the pharmacokinetic mechanism of diuretic action in CHF could be affected by the altered pharmacokinetic mechanisms with concomitant liver diseases. This holds true especially in diuretics that are albumin bound, like furosemide. The decreased protein synthesis in liver diseases can have impaired pharmacokinetic effects on the action of diuretics like furosemide used in the management of heart failure exacerbations.

We also take into account the simple confounding factor in our study, which is that two pathologic conditions existing together could be more harmful than a single pathologic condition. Paik et al. [20] performed an analysis based on the data from National Health Statistics from the years 2007 to 2017 and noted an increasing trend in mortality associated with chronic liver conditions, especially cirrhosis, and attributed it mainly to the risk of hepatocellular carcinoma in cirrhosis patients. To overcome this limitation, we matched the samples of heart failure admissions using a propensity scoring system, where we matched the samples into ones with and without liver diseases, while all other healthcare and population characteristics remained the same, and the cohort with concomitant liver failure/cirrhosis had a significantly higher in-hospital mortality. Revisiting the study by Paik et al. [20], it was also noted that irrespective of the occurrence of hepatocellular carcinoma, alcoholic liver diseases tend to have a worse morbidity/mortality effect than non-alcoholic/metabolic diseases of the liver, which falls in line with our crude comparison showing worse outcomes for ADHF patients with alcoholic cirrhosis when compared to non-alcoholic/metabolic diseases of the liver. While considering the metabolic diseases of the liver, the association with cardiac dysfunction could be explained by the etiologic similarities, especially the controversial metabolic syndrome X. Kim et al. [21] based on an analysis of 77,671 patients sorted from the Third National Health and Nutrition Examination Survey, found that nonalcoholic liver diseases and metabolic conditions are part of a wider spectrum of disorders, especially hypertension and diabetes mellitus, which share etiological similarities with that of systolic as well diastolic heart failure.

Analyzing the results of our study, another fact that is worthy of mentioning was that the adverse mortality and morbidity outcomes had a correlation with the age of the admissions. Valbusa F et al. performed [22] a retrospective analysis where it was noted that age of the admissions had a positive correlative effect on the effect of hepatic fibrosis score and 30-day mortality in heart failure admissions. One factor that we could not account for based on the nature of the database was conditions like hepatic vein thrombosis that could make the liver more prone to ischemic injury, especially in patients with heart failure [23]. Another potential confounder, which we addressed in both the multivariate regression analysis as well as the propensity score matching, was alcohol use disorder. Alcohol abuse has a significant impact on patients with both heart failure and CLD. More affluent neighborhoods are associated with better health outcomes, while disadvantaged areas often have worse outcomes [10]. Programs targeting disadvantaged populations through outreach and financial assistance could be key in mitigating this effect [24]. Additionally, ALD and NAFLD are more prevalent in certain racial and ethnic groups, which could exacerbate disparities in outcomes [25,26]. Studies have also demonstrated that racial and ethnic minorities, including Black, Hispanic, and Indigenous American/Pacific Islander patients, tend to have longer hospital stays without corresponding improvements in care quality, further polarizing disparities in outcomes [27,28]. Moreover, ethnic minorities with liver disease are at a greater risk of being uninsured, which could further explain disparities [29,30]. This presents a potential avenue for future research within our study to explore these disparities in more detail.

Looking at the results of our study with the insights from the literature review, the adverse outcomes of heart failure exacerbations in admissions with concomitant liver diseases could be explained both by the intrinsic molecular and physiologic effects liver dysfunction can have on myocardium, volume imbalances, and medication non-adherence. Liver pathologies, especially alcohol-related ones, can be reversed by abstinence from the same, which is known to preserve/recover synthetic functions of the liver. This could have positive impacts on the physiologic and molecular effects liver dysfunction will have on ventricular remodeling. Clinicians can emphasize this fact, especially in patients with alcoholic liver diseases, which has been proven to have a more deleterious effect on cardiac function when compared to non-alcoholic liver diseases. Volume status and medication adherence go together, and splanchnic vasodilation plays a key role in both. Effects of splanchnic vasodilation can be mitigated by portal hypertension, and advanced modalities in the treatment of portal hypertension, especially TIPS, etc., theoretically would have superior effects in heart failure patients. This would serve as a potential field for future research. Lastly, the socio-economic and racial factors would be an avenue for public health and preventative medicine to intervene upon.

## 5. Conclusions

Based on the results of our statistical analysis, among admissions with ADHF, coexisting diagnosis of CLD was associated with higher mortality, mean length of hospital stay, and mean of total charges. The mortality and healthcare resource utilization parameters were even higher for ADH admissions with a comorbid diagnosis of alcoholic liver disease. To avoid confounding from comorbid diagnosis, age, and genetic factors, we performed a multivariate logistic regression that also proved a significant association between diagnosis of CLD/alcoholic liver disease and all-cause mortality in ADHF admissions. After matching cohorts with and without diagnosis of CLD selected from the ADHF population, we found that the cohort with a comorbid diagnosis of CLD among ADHF admissions had a significantly higher proportion of in-hospital mortality when compared to the cohort without CLD selected from ADHF admissions. No significant healthcare or social disparities were seen between the general ADHF admissions versus the ones with a comorbid diagnosis of CLD.

## Figures and Tables

**Table 1 medsci-13-00019-t001:** Population characteristics of CHF admissions with chronic liver disease and alcoholic cirrhosis.

Population Characteristics of Congestive Heart Failure Admissions	With Chronic Liver Disease	With Alcoholic Cirrhosis
Mean Age	63.36 (64.16–64.56)	61.9 (61.69–62.17)
Sex		
Male	67.45%	73.62%
Female	32.55%	26.38%
Race		
Whites	66.7%	65.6%
Blacks	15.97%	15.35%
Hispanics	11.72%	13.7%
Median Household Income		
Lower quartiles	59.47%	61.21%
Upper quartiles	40.53%	38.8%
Charlson comorbidity index score > 5	54%	54.87%

**Table 2 medsci-13-00019-t002:** Analyzing in-hospital outcomes of CHF admissions with CLD and alcoholic cirrhosis.

	Proportion DIED	Mean Length of Stay (Days)	Mean of Total Charges
Congestive heart failure admissions without the diagnosis of CLD	0.045 (0.044–0.046)	6.18 (6.13–6.23)	USD 79,946.21 (77,644.87–82,247.54)
CHF admissions with coexisting CLD	0.054 (0.053–0.057)	6.95 (6.84–7.06)	USD 88,068.17 (85,201.03–90,935.3)
CHF admissions with coexisting alcoholic liver disease	0.071 (0.067–0.076)	7.26 (7.09–7.43)	USD 88,161.37 (84,764.61–91,558.13)

**Table 3 medsci-13-00019-t003:** Association between coexisting diagnosis of CLD and all-cause mortality among ADHF admissions.

Association Between All-Cause Mortality in CHF Admissions	OR	95% Confidence Interval	*p*-Value
CLD: unadjusted odds ratio	1.39	1.31–1.46	0.00
CLD: adjusted for comorbidities, age, race, and sex	1.23	1.17–1.29	0.00

**Table 4 medsci-13-00019-t004:** Association between all-cause mortality and diagnosis of alcoholic liver disease in ADHF admissions.

All-Cause Mortality in CHF Admissions	OR	95% Confidence Interval	*p*-Value
Alcoholic liver disease: unadjusted odds ratio	1.82	1.69–1.96	0.00
Alcoholic liver disease: adjusted for comorbidities, age, race, and sex	1.63	1.51–1.74	0.00

**Table 5 medsci-13-00019-t005:** Propensity score and nearest neighbor matching sample analysis on ADHF admissions. CLD +: admissions with documented CLD. CLD −: admissions without documented CLD.

Propensity-Matched Cohorts	Number of Observations	Mean Age	Prevalence of Smoking	Prevalence of Alcohol Abuse	Prevalence of CKD	Proportion Died
CLD + cohort	3973	42.41	0.0078	0.163	0.23	0.042 (0.036–0.049)
CLD − cohort	3973	42.40	0.0078	0.163	0.23	0.027 (0.022–0.033)

## Data Availability

The data presented in this study are openly available in NIS database at https://hcup-us.ahrq.gov/db/nation/nis/nisdbdocumentation.jsp (accessed on 19 January 2025).

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
