# Peer review of "Outcomes in Acute Decompensated Congestive Heart Failure Admissions with Chronic Liver Disease: A Nationwide Analysis Using the National Inpatient Sample"

_medsci, 2025, doi:10.3390/medsci13010019_

Round 1
Reviewer 1 Report
Comments and Suggestions for Authors
Dear Editor of Medical Sciences - MDPI Thank you for inviting me to be a Referee for this scientific article: ‘OUTCOMES IN CONGESTIVE HEART FAILURE ADMISSIONS WITH CHRONIC LIVER DISEASE: A NATIONWIDE ANALYSIS USING THE NATIONAL IN-PATIENT SAMPLE”. Although I have transcribed my opinion of what is requested in the Medical Sciences - MDPI electronic form, I would like to add some comments. - There are also some aspects that deserve to be changed, which are referenced in red: (in red proposals to be potentially changed and also to be praised).
Title: why not include “Acute Decompensated Congestive Heart Failure” Outcomes in Chronic Liver Disease”, as in the Abstract (why include “CONGESTIVE HEART FAILURE ADMISSIONS” in the title?
Regarding the identification of the authors:
- Vivek Joseph Varughese[1], Vignesh Krishnan Nagesh[2],…
Is it Vivek Joseph Varughese[1] or should it be Vivek Joseph Varughese1,,,? [2]2--- The email addresses of the various authors are also missing.
ABSTRACT
Well written, and provides a correct and adequate synthesis of the results found.
INTRODUCTION
Generally well written and with the necessary topics, well covered, but:
-Line? “…renin-angiotensin system inhibitors, evidence-based β-blockers, mineralocorticoid inhibitors and sodium glucose cotransporter 2 inhibitors.(1)…”; renin-angiotensin system inhibitors (ACE inhibitors or ARB) or ARNI…. evidence-based β-blockers, mineralocorticoid inhibitors and sodium glucose cotransporter 2 inhibitors [1].
From now on, all citation numbers must be revised: instead of ( )…it should be [ ].
- Lines?: “Volume overload in heart failure is an end-common pathway of a diverse number of disease states, including congenital heart disease, ischemia, arrhythmia, and external factors.”; …should read ischemic coronary disease, …external factors (which for example?)..
- Line: “Complex physiological factors determine the timing of the systolic function reduction”; only physiological factors or also and above all structural factors such as….
- Lines?: “While volume overload is more generalized secondary to increased intracardiac filling pressures in CHF, the pathophysiology of volume overload in CLD is more complex.”; in what sense and what are these pathophysiological changes different from those seen in ADHF? (otherwise the text will be too generic…)
- Lines?: “the primary driver remains volume overload due to portal hypertension.”; just portal hypertension or also hypoalbuminemia and reduced oncotic pressure?
- Lines?: Systemic complications of portal hypertension are varied, and includes esophageal and related varices, portal gastropathy, and splenomegaly”; it can read …are varied and includes. esophageal varices, portal hypertensive gastropathy and colopathy, hepatic encephalopathy, ascites, hepatorenal syndrome and splenomegaly.
METHODS
The selected diagnoses cannot be criticized...but they are very generic diagnoses that could be better scrutinized and sub-analyzed in terms of their criteria and parameters and subgroups (to draw better conclusions about the results obtained). Like this:
- “We identified 4391904 patients who were admitted with the diagnosis of congestive heart failure. In this population sample, 146050 people had coexisting unspecified chronic liver disease and 59260 of the admissions had a diagnosis of alcoholic cirrhosis.”; congestive heart failure or Acute Decompensated Heart Failure (ADHF?) , because it is possible to be hospitalized for several other reasons and have associated CHF (and not ADHF);
-Is it also possible in this context to make it clearer which diagnostic criteria are used in the NIS on ADHF? (necessarily very briefly, although the text of this article refers to the NIS database); Also in this context, what type of chronic liver disease was included in this sample: chronic persistent hepatitis or chronic active hepatitis or cirrhosis? Non-Alcoholic Fat Disease or Alcoholic Fat Disease? What criteria are used to consider a definitive diagnosis of alcoholic liver cirrhosis?; what other etiologies for chronic liver disease (non-alcoholic fatty disease? hepatitis B or C, or hemochromatosis, or Wilson's disease?....or biliary cirrhosis...in cases of liver cirrhosis...what sensitivity and/or specificity do the criteria used in ICD-10 in the diagnosis of chronic liver disease and/or alcoholic cirrhosis (see for example: Hayward KL, Johnson AL, Mckillen BJ, et al. ICD-10- AM codes for cirrhosis and related complications: key performance considerations for population and healthcare studies. BMJ Open Gastro 2020;7:e000485. ? (serum albumin, bilirubin and prothrombin time; presence or absence of ascites and encephalopathy) as these different parameters can have a very different prognostic impact on patients admitted for ADHF... some sub-analysis was carried out regarding the correlation with these different parameters related to the Child-Pugh classification).
RESULTS
The results, although correct, can be better presented.
Lines?: “We identified 4391904 patients who were admitted with the diagnosis of congestive heart failure. In this population sample, 146,050 people had coexisting unspecified chronic liver disease and 59,260 of the admissions had a diagnosis of alcoholic cirrhosis; …146050 people had coexisting unspecified chronic liver disease (which chronic liver disease did these patients have – see review above…..what is the correct diagnosis…cirrhosis??? Active chronic hepatitis…this aspect is important because the chronic inflammatory repercussion and the syndromes that accompany chronic liver disease, as very well stated in the Introduction) will certainly have a very different impact on the prognosis and costs and the length of stay in patients admitted due to ADHF (these aspects should be clearly discussed further.
Several texts such as the following can be found easily and some of their content can be used later in the global discussion:
“Peripheral and splanchnic vasodilation … Initially, a reduction in systemic vascular resistance is compensated by an increase in cardiac output, and effective arterial blood volume remains in the normal range. In advanced stages of cirrhosis, a marked reduction in systemic vascular resistance cannot be compensated by a further increase in cardiac output, and this leads to underfilling of arterial circulation. At this stage, there is activation of vasoconstrictor systems such as renin-angiotensin, sympathetic nervous system, and antidiuretic hormone, which maintain effective arterial blood volume and arterial pressure [23]. On the other hand, these compensatory systems are the leading cause of sodium and water retention that lead to ascites formation with the disease progression [24]. Furthermore, a prolonged activation of the aforementioned vasoconstrictor systems leads to severe renal vasoconstriction and reduced glomerular filtration rate, a condition that may escalate into a progressive renal insufficiency, namely HRS…
"cirrhotic cardiomyopathy" describes impaired contractile responsiveness to stress, diastolic dysfunction and electrophysiological abnormalities in patients with cirrhosis without known cardiac disease. Underlying circulatory and cardiac dysfunctions are the main determinant in the development of hepatorenal syndrome in advanced cirrhosis… portopulmonary syndrome, hepatopulmonary syndrome, and systemic hyperdynamic circulation (see later)… as well as mechanisms involved (cytokines, NO; to be discussed).
- Line?: “In the subpopulation with coexisting ADHF and chronic liver disease, mean age was 64,364(64,164 – 64,565). 66.7% were white, 15.97% were black and 11.72% were Hispanic”; why in Table 1 the results are expressed in hundredths places and not decimals as in the present text which is in mellisimal places?; 64.364v instead of 64.4?. throughout the text it is better to express the results in numbers and only one decimal place
- Line?: “Analyzing the charlson comorbidity index, 53.99% of the population had a charlson comorbidity”; It should be: Analyzing the Charlson comorbidity index, 54.0% of the population had a Charlson comorbidity
- Line?: “Considering the median household income quartile…”; What median value was estimated?
- Line?: It is not clear whether the population with coexisting ADHF and chronic liver disease has a higher or lower Charlson index than the population with only ADHF and no chronic liver disease and/or alcoholic cirrhosis....?" and whether there are statistically significant differences ( the p value is missing in the results);
- Line?: “Analyzing the subpopulation of ADHF admissions with alcoholic cirrhosis of the liver, mean age of the population was 61.9 (61.69 62.17). 65.6% were white, 15.35% were black and 13.7% were Hispanic. Tabulating the charlson comorbidity index, 54.9% of the population had a charlson comorbidity”; but there is a lack of statistical treatment that compares the results of patients with ADHF with chronic liver disease/cirrhosis and without chronic liver disease/cirrhosis in the evaluated parameters (mortality, length of stay and hospitalization costs). See for example the type of Tables published by Khalid et al (2020) citation [11] from the authors of the current work columns p and captions.
- The statistical treatment of the data in Table 2 is missing (as correctly expressed in Tables 3 and 4.
In Tables 3 and 4, why is the column referring to the value of p in the 3rd column and not in the 4th column? Next to the 95% confidence interval column?
- Line?: “The proportion of mortality was even higher in ADHF admissions with comorbid diagnosis of alcoholic liver disease: 0.071 (0.067 – 0.076).” try to explain this significant difference in the discussion…the patients also had alcoholic cardiomyopathy and what is the LV Ejection Fraction of these patients compared to those who did not have alcoholic cirrhosis?
Line?: “and diabetes. (Exact matching was done for Age, race, smoking, alcohol abuse and CKD). Probit…”; must be exact…age.
DISCUSSION
The discussion chapter has to be generally improved, I would even say that it has to be completely reformulated. There are several points that need to be improved, to enrich and broaden the scope of the results found in this study; in many segments it is confusing and seems disconnected from the population included in this study. It also includes definitions that are not always correct. The discussion suffers from an important error, that is, it always confuses NAFLD with cirrhosis…
Line?: where it says “2016 by Khalid et al. [11], looking at outcomes of heart failure admissions in patients with cirrhosis, looking at a sample of 355,455 admissions for ADHF, with 3.4 % mortality in the cohort with documented cirrhosis, significantly higher than the general mortality of heart failure”; should be added/modified to …propensity-matched and multivariate logistic regression analysis of 7595 patients with both ADHF and cirrhosis showed that mortality was almost two times higher in patients with ADHF and cirrhosis compared to patients without cirrhosis (p<0.001), mean hospital length of stay was 1.2 times higher in patients with ADHF and cirrhosis (p\0.001), but the mean total burden of hospitalization was not statistically significant.”
- Line?: “looked at whether alcoholic cirrhosis holds a more potential adverse effect on heart failure outcomes when compared to general chronic liver diseases. This was based on the pathological difference alcoholic cirrhosis can have when compared to general CLD [13”]; but what are these pathological differences…myocardial depression and structural changes induced by ethanol in the myocardium? Discuss this aspect better and better explain the potential pathophysiological differences…what mechanisms?
Line?: it reads “This was based on the pathological difference alcoholic cirrhosis can have when compared to general CLD [13]”; but in the study by Toshikun et al (2014) cited as [13] what is compared are the various changes found “…differences between alcoholic liver disease and nonalcoholic fatty liver disease” and non-alcoholic cirrhosis and CLD.
- Lines?: it reads “As mentioned in the introduction section, CLD playing a role in medication non-adherence, especially ACE/ARBs in the management of systolic heart failure, there has been several studies looking into pathophysiologic mechanisms of cirrhosis as well as Non-Alcoholic Fatty Liver Diseases (NAFLD) having negative impact on the left ventricular function, adding further molecular level evidence for the findings we have in our study.”; this sentence is very confusing and We don't understand the interconnection that the authors want to make...it must be completely modified and give it a logical sequence; only to ACEi/ARB???; what molecular level evidence? (describe related molecular changes…).
Lines?: reads “. In the literature review by Mantovni et al. [14], it was noted that adult individuals with NAFLD in the absence of severe obesity, hypertension, and diabetes have mildly increased left ventricular mass and early features of diastolic dysfunction”; but in this article by Mantovani et al (2016) you can read Spectrum of the most important cardiac complications associated with NAFLD: Coronary heart disease, cardiac hypertrophy, heart failure, atrial fibrillation, aortic valve sclerosis, mitral annulus calcification, QTc prolongation”, ,,; Therefore, the text must be completed with the spectrum of diseases associated with NAFLD, correcting to Mantovani (instead of Mantovni); It is also not understood what the relationship between NAFLD is and the patients included in this study as in the methods chapter there is no reference to the number of patients included in the study with the aforementioned pathology (NAFLD.)
- Lines?: it reads “…world study by Nan Hee Kim et al [15], 1886 patients with CT proven cirrhosis, and no known cardiovascular risk factors were selected from the Korean Genome Registry. “; but contrary to what the authors claim, it was not “CT proves cirrhosis” but rather NAFLD diagnosed by CT “The liver attenuation index (LAI), derived from the difference between the mean hepatic and splenic attenuation, was used as a parameter to diagnose NAFLD ..,.The participants were divided into four groups, based on the presence of NAFLD, metabolic syndrome (MetS), neither or both. NAFLD was diagnosed by CT…”; It should also be written only Kim et al and not Nan Hee Kim et al.
- Lines?: it reads “It was noted that presence of cirrhosis was associated with an elevated E/Ea ratio with decreased TDI Ea velocity in these patients with statistical significance.”; But what is the meaning of these aforementioned changes? indicate diastolic dysfunction associated with greater morbidity and mortality (explain better---" The mitral E/Ea ratio as an index of LV diastolic filling pressure ...see Kim et al); and once again the study cited addresses patients with NFALD “Non-alcoholic fatty liver disease, metabolic syndrome and subclinical cardiovascular changes in the general population”
- Lines?: it reads “The pathological effects of Liver cirrhosis on cardiac musculature were analyzed further in a real-world study by VanWagner et al. [16],”; again the confusion between cirrhosis and NAFLD... NAFLD can evolve into cirrhosis but they are not the same nosological entities... according, for example, to Pouwels et al. (Non-alcoholic fatty liver disease (NAFLD): a review of pathophysiology, clinical management and effects of weight loss BMC Endocrine Disorders. 2022; 22:63) “Non-alcoholic fatty liver disease (NAFLD) is a common cause of chronic liver disease worldwide. NAFLD is a spectrum of the disease characterized by hepatic steatosis when no other causes for secondary hepatic fat accumulation (e.g., excessive alcohol consumption) can be identified. NAFLD ranges from the most benign condition of non-alcoholic fatty liver (NAFL) to non-alcoholic steato hepatitis (NASH), which is at the most severe end of the spectrum. NAFLD may progress to fibrosis and cirrhosis [1, 2].
In NAFLD, hepatic steatosis is present without evidence of inflammation, whereas in NASH, hepatic steatosis is associated with lobular inflammation and apoptosis that can lead to fibrosis and cirrhosis”.
- Lines?: it reads “The pathological effects of Liver cirrhosis on cardiac musculature were analyzed further in a real-world study by VanWagner et al. [16], it was noted that patients with NAFLD had lower early diastolic relaxation(e’) velocity (10.8 +/- 2.6 Vs 11.9 +/…”; but cirrhosis or NAFLD…
- Lines: reads “and lower Global Longitudinal strain pattern. The study had also noted subclinical cardiac remodeling in these patients.”; What does it indicate and what is the implication lower?? Global strain and also that subclinical remodeling changes were found?
-Lines: it reads “beyond the clinical aspect of CLD being an important factor for medication non-adherence.”; What does one thing have to do with the other…what is the relationship with non-adherence to therapy?
-Lines: reads “This also provides an explanation at the molecular level for the findings we have in our study.”; molecular levels? What does this mean?
Lines: it reads “Beyond interactions at the molecular level, the etiologic association of tachyarrhythmia induced cardiomyopathies and liver diseases is yet another factor that could have a harmful impact in heart failure management. In a retrospective analysis by Ballestri et al., [17] it was noted that NAFLD itself, especially in its more severe forms, exacerbates systemic/hepatic insulin resistance, causing atherogenic dyslipidemia, and releases a variety of pro-inflammatory, pro-coagulant and pro-fibro genic medications that may play important roles in the pathophysiology of cardiac and arrhythmic complications.” The logic of this sequence in the discussion is not understood…separate truths but without interconnection…
- Lines: it reads “…medication nonadherence remains the leading cause of heart failure exacerbations”; this is not true…many other precipitating factors contribute more to ADHF (infections, arrhythmias, myocardial ischemia, pulmonary embolism, excess food….thyroid dysfunction….)…to discuss further and the relationship between non-adherence and CLD remains unclear. ???.
Lines: it reads “Robert W. Shrier [18], it was noted that third spacing/volume overload in both cardiac and liver failure is largely driven by the non-osmotic activation”; Shrier (not Robert W. Shrier [); What does non-osmotic activation mean?
- Lines: reads “while decrease in systemic vascular resistance…”; while activated by the decrease?...non natriuretic doses??; relationship hyperkaemkia and cardiac dysfunction???
- Lines: it reads “the effect of diuretics in cirrhosis and cardiac failure is mainly mediated by the pharmacodynamics, while in conditions like CKD, its mediated by the pharmacokinetic effect [19].”; It is not understood and what does this have to do with the results found in the study?
- Lines: It reads “especially the controversial syndrome X. Kim et al. [21]. “Syndrome X??? What syndrome is this coronary syndrome X????
- Lines: reads “Kim et al. [21]. Based on their analysis of 77,671 patients sorted from the Third National Health and Nutrition Examination Survey,” ; Kim et al “We analyzed data from 7,761 participants in the Third National Health and Nutrition Examination Survey and their linked mortality through 2015. NAFLD was diagnosed by ultrasonographic evidence of hepatic steatosis without other known liver diseases”; 7761 and not 77671…NAFLD was diagnosed….
- Valbusa F…; Valbusa et al.
- Lines: it reads “studied 212 elderly patients who were consecutively admitted with acute HF to the Hospital; retrospective study?
- Lines: it reads “One factor that we could not account for based on the nature??? of the database were conditions like hepatic vein thrombosis that could make the liver more prone to ischemic injury, especially in patients with heart failure.[23]
- Lines: it reads “More affluent neighborhoods are associated with better health outcomes, while disadvantaged areas often have worse outcomes11???”. What does this have to do with the current study? (no results are mentioned in this regard)…
- Lines: what is the relationship between the following content and the current study???? And on top of that without objective data…average ethanol consumption g/day….for example…average annual income….???
It reads “More affluent neighborhoods are associated with better health outcomes, while disadvantaged areas often have worse outcomes11. Programs targeting disadvantaged populations through outreach and financial assistance could be key in mitigating this effect [24]. Additionally, ALD and NAFLD are more prevalent in certain racial and ethnic groups which could exacerbate disparities in outcomes[25,26].Studies have also demonstrated that racial and ethnic minorities, including Black, Hispanic, and Indigenous American/Pacific Islander patients, tend to have longer hospital stays without corresponding improvements in care quality, further polarizing disparities in outcomes [27,28]. Furthermore, ethnic minorities with liver disease are at a greater risk for being uninsured, which could further explain disparities [29,30]”.
- Lines: it reads “Concluding,…but it is outside the Conclusions chapter
- Lines: it reads “could be explained both by the intrinsic molecular and physiological effects liver dysfunction can have on myocardium, volume imbalances as well as medication non-adherence.”; ….intrinsic molecular and physiological effects liver dysfunction (which ones? Molecular and pathophysiological) (take advantage, for example, of parts of the text published by Botros Shenoda and Joseph Boselli (Vascular syndromes in liver cirrhosis Clin J Gastroenterol. . 2019 Oct;12(5): 387-397); “Important vascular features encountered in liver disease include portal hypertension, splanchnic overflow, abnormal angiogenesis and shunts, portopulmonary syndrome, hepatopulmonary syndrome, and systemic hyperdynamic circulation. Redistribution of effective circulatory volume deviating from vital organs and pooling in splanchnic circulation is also encountered in liver patients which may lead to devastating outcomes as hepatorenal syndrome. phenomena and vascular dysfunction in one system may lead to the development of another in a different system…”.
And also take advantage of aspects present in these excerpts from the excellent discussion of the publication by Khalid et [11]: “In: Khalid In-Hospital Outcomes of Patients with Acute Decompensated Heart Failure and Cirrhosis: An Analysis of the National Inpatient Sample Yaser S. Khalid” Both disease processes instigate arterial underfilling, which in turn stimulates a sympathetic response that releases non-osmotic vasopressin. This results in an increased sympathetic tone and renal adrenergic stimulation, and, as a result of this change in tonicity, activates the renin–an giotensin aldosterone system (RAAS) [11–13]. Ultimately, sodium and water retention occurs as a result of these processes… and presents as hypertension, pulmonary or peripheral edema, continued or worsening HF, ascites, HRS, and cardiorenal syndrome…
- Can the authors discuss the involvement of the chronic inflammatory process present in CLD, circulating cytokines, the NO pathway and other markers/factors that affect the heart, affecting systolic and diastolic function and that certainly explain and aggravate ADHF and its poor prognosis when there is concomitant CLD (in porto-pulmonary syndrome, cardio-renal syndrome, cirrhotic cardiomyopathy, hydrothorax related to CLD)…( "cirrhotic cardiomyopathy" - cardiovascular complications following these procedures are common, with pulmonary edema being the most common complication. Other complications include overt heart failure, arrhythmia, pulmonary hypertension, pericardial effusion, and cardiac thrombus formation.). In: Fede et al. 2015. Cardiovascular dysfunction in patients with liver cirrhosis ADHF admissions with alcoholic liver disease…);
-Lines: reads “This could have positive impacts on the physiological and molecular effects liver dysfunction will have on ventricular remodeling”; very generic and vague phrase…but what physiological and molecular alterations?
What remains to be discussed or improved in the Discussion/Results/Methods:
- Discuss better, as already stated, the pathophysiology of cirrhosis and ADHF… which will even contribute to the worst results found;
- It remains to indicate the diagnostic criteria for ADHF and chronic liver disease and the subtypes of CLD (the number of patients with NAFDL, steatohepatitis, true cirrhosis); stages/Child-Pugh classification of cirrhosis and potential correlation of hypoalbuminemia with ADHF/edema and outcomes.
- How many patients with ADHF met criteria for Hepatorenal Syndrome (take the opportunity to discuss some pathophysiological mechanisms)
Conclusions
- It reads “Based on the results of our statistical analysis,…”; Could…if it weren’t like this???? Change the sentence
- It reads “comorbid diagnosis, age, and genetic factors,”…which genetic factors were studied?
REFERENCES
Adequate, but:
Citations must be revised in their format (correct the aspects highlighted in red in citation 1 and extend these corrections to all citations; correct other aspects highlighted)
1. Patel J, Rassekh N, Fonarow GC, Deedwania P, Sheikh FH, Ahmed A, Lam PH. Guideline Directed Medical Therapy for the Treatment of Heart Failure with Reduced Ejection Fraction. Drugs. 2023 Jun;83(9):747-759.
Should be: Patel, J.; Rassekh, N.; Fonarow, G. C.; Deedwania, P.; Sheikh, F.H.; Ahmed, A.; Lam, P. H. Guideline Directed Medical Therapy for the Treatment of Heart Failure with Reduced Ejection Fraction. Drugs. 2023 Jun (remove publication month);83(9):747-759.
2. Goldgrab D, Balakumaran K, Kim MJ, Tabtabai SR. Updates in heart failure 30-day readmission prevention. Heart Fail Rev. 2019 Mar;24(2):177-187. doi: 10.1007/s10741 018-9754-4. PMID: 30488242.
Should be: Goldgrab, D.; Balakumaran, K.; Kim, M.J.; Tabtabai, S.R. Updates in heart failure 30-day readmission prevention. Heart Fail Rev. 2019 Mar (remove publication month);24(2):177-187. doi: 10.1007/s10741 018-9754-4. PMID: 30488242.
Because in some citations it uses 10 or more authors and in others only 3 of more than 10 and they include et al, for example (the number of authors cited must be standardized)
Kim NH, Park J, Kim SH, et al: Non-alcoholic fatty liver disease, metabolic syndrome and subclinical cardiovascular changes in the general population Heart 2014;100:938-943.
Kim NH, Park J, Kim SH, ……Nan Hee Kim, Juri Park, Seong Hwan Kim, Yong Hyun Kim, Dong Hyuk Kim, Goo-Yeong Cho, Inkyung Baik, Hong Euy Lim, Eung Ju Kim, Jin Oh Na, Jung Bok Lee, Seung Ku Lee, Chol Shin
19. Brater C. Update in Diuretic Therapy: Clinical Pharmacology. Seminars….
20. Paik JM, Golabi P, Biswas R, Alqahtani S, Venkatesan C, Younossi ZM. Nonalcoholic Fatty Liver Disease and Alcoholic Liver Disease are Major Drivers of Liver Mortality in the United States. Hepatol Commun. 2020;4(6):890-903. Published ???????2020 Apr 4. doi:10.1002/hep4.1510
22. Valbusa F, Bonapace S, Agnoletti D, et al. Nonalcoholic fatty liver disease and increased risk of 1-year all-cause and cardiac hospital readmissions in elderly patients admitted for acute heart failure. PLoS One. 2017;12(3):e0173398. Published??? 2017 Mar 13. doi:10.1371/journal.pone.0173398
Filippo Valbusa 1, Stefano Bonapace 2, Davide Agnoletti 1, Luca Scala 1, Cristina Grillo 1, Pietro Arduini 3, Emanuela Turcato 3, Alessandro Mantovani 4, Giacomo Zoppini 4, Guido Arcaro 1, Christopher Byrne 5 6, Giovanni Targher 4 (quote all authors?)

Author Response
why not include “Acute Decompensated Congestive Heart Failure” Outcomes in Chronic Liver Disease”, as in the Abstract (why include “CONGESTIVE HEART FAILURE ADMISSIONS” in the title?
Response: Admissions for decompensated heart failure (acute): ICD 10 codes I150 * were used to select the index admission population so as to exclude admissions with comorbid CHF but not the cause of admission ( will change the text to specify this)
Is it also possible in this context to make it clearer which diagnostic criteria are used in the NIS on ADHF? (necessarily very briefly, although the text of this article refers to the NIS database); Also in this context, what type of chronic liver disease was included in this sample: chronic persistent hepatitis or chronic active hepatitis or cirrhosis? Non-Alcoholic Fat Disease or Alcoholic Fat Disease? What criteria are used to consider a definitive diagnosis of alcoholic liver cirrhosis?; what other etiologies for chronic liver disease (non-alcoholic fatty disease? hepatitis B or C, or hemochromatosis, or Wilson's disease?....or biliary cirrhosis...in cases of liver cirrhosis...what sensitivity and/or specificity do the criteria used in ICD-10 in the diagnosis of chronic liver disease and/or alcoholic cirrhosis (see for example: Hayward KL, Johnson AL, Mckillen BJ, et al. ICD-10- AM codes for cirrhosis and related complications: key performance considerations for population and healthcare studies. BMJ Open Gastro 2020;7:e000485. ? (serum albumin, bilirubin and prothrombin time; presence or absence of ascites and encephalopathy) as these different parameters can have a very different prognostic impact on patients admitted for ADHF... some sub-analysis was carried out regarding the correlation with these different parameters related to the Child-Pugh classification).
Response: The DXCCR elixihhauser software was used to select admissions with documented CLD. This was done so as to avoid the confounding factor of whether the diagnosis of CLD was present on admission / a chronic resolved condition. The software only includes CLD and alcoholic liver disease, hence NAFLD and hepatitis related cirrhosis were excluded in the stratified analysis as these diagnoses and related stats would be subject to confounding
146050 people had coexisting unspecified chronic liver disease (which chronic liver disease did these patients have – see review above…..what is the correct diagnosis…cirrhosis??? Active chronic hepatitis…this aspect is important because the chronic inflammatory repercussion and the syndromes that accompany chronic liver disease, as very well stated in the Introduction) will certainly have a very different impact on the prognosis and costs and the length of stay in patients admitted due to ADHF (these aspects should be clearly discussed further.
Response: same reason as above. Using the DXCSSR software, only CLD and alcoholic cirrhosis were documented and other causes of CLD, if included in the stratification and analysis would be prone to confounding
congestive heart failure or Acute Decompensated Heart Failure (ADHF?) , because it is possible to be hospitalized for several other reasons and have associated CHF (and not ADHF);
Response: admissions having ICD codes documented for ADHF as the main diagnosis ( I10_DX1) in the NIS database was used in the analysis to exclude admissions having CHF as chronic comorbidity but not as the cause of admission
-Is it also possible in this context to make it clearer which diagnostic criteria are used in the NIS on ADHF? (necessarily very briefly, although the text of this article refers to the NIS database); Also in this context, what type of chronic liver disease was included in this sample: chronic persistent hepatitis or chronic active hepatitis or cirrhosis? Non-Alcoholic Fat Disease or Alcoholic Fat Disease? What criteria are used to consider a definitive diagnosis of alcoholic liver cirrhosis?; what other etiologies for chronic liver disease (non-alcoholic fatty disease? hepatitis B or C, or hemochromatosis, or Wilson's disease?....or biliary cirrhosis...in cases of liver cirrhosis...what sensitivity and/or specificity do the criteria used in ICD-10 in the diagnosis of chronic liver disease and/or alcoholic cirrhosis (see for example: Hayward KL, Johnson AL, Mckillen BJ, et al. ICD-10- AM codes for cirrhosis and related complications: key performance considerations for population and healthcare studies. BMJ Open Gastro 2020;7:e000485. ? (serum albumin, bilirubin and prothrombin time; presence or absence of ascites and encephalopathy) as these different parameters can have a very different prognostic impact on patients admitted for ADHF... some sub-analysis was carried out regarding the correlation with these different parameters related to the Child-Pugh classification).
Response: totally agree with the suggestion. However, it remains a limitation of the NIS database of not giving longitudinal data for the diagnosis. Ex, a diagnosis of cirrhosis selected with appropariate ICD codes for the same in the admission population and related analysis could be confounded by the fact of not knowing whether this was a recent diagnosis or one made in the remote past. To avoid this, chronic medical conditions that are ongoing issues for the patient was selected using the DXCSSR software, which only documented CLD and alcolic cirrhosis, hence only used these two in the analysis
Line?: “In the subpopulation with coexisting ADHF and chronic liver disease, mean age was 64,364(64,164 – 64,565). 66.7% were white, 15.97% were black and 11.72% were Hispanic”; why in Table 1 the results are expressed in hundredths places and not decimals as in the present text which is in mellisimal places?; 64.364v instead of 64.4?. throughout the text it is better to express the results in numbers and only one decimal place
Response: will change and use rounded places
What median value was estimated?
Response: database calculates the median interquartile range for the admissions based on the household income of the neighborhood based on the zip codes used at admission: this has been added to the text
Line?: It is not clear whether the population with coexisting ADHF and chronic liver disease has a higher or lower Charlson index than the population with only ADHF and no chronic liver disease and/or alcoholic cirrhosis....?" and whether there are statistically significant differences ( the p value is missing in the results);
Response: only general proportion in each charleston index group was mentioned and did not include the 95% intervals to assess statistical difference among groups .
Line?: “Analyzing the subpopulation of ADHF admissions with alcoholic cirrhosis of the liver, mean age of the population was 61.9 (61.69 62.17). 65.6% were white, 15.35% were black and 13.7% were Hispanic. Tabulating the charlson comorbidity index, 54.9% of the population had a charlson comorbidity”; but there is a lack of statistical treatment that compares the results of patients with ADHF with chronic liver disease/cirrhosis and without chronic liver disease/cirrhosis in the evaluated parameters (mortality, length of stay and hospitalization costs). See for example the type of Tables published by Khalid et al (2020) citation [11] from the authors of the current work columns p and captions.
response: since we were not able to statistically differentiate the admissions based on the charleston comorbidity index, avoiding comparisons based on the same among admission groups
Line?: “The proportion of mortality was even higher in ADHF admissions with comorbid diagnosis of alcoholic liver disease: 0.071 (0.067 – 0.076).” try to explain this significant difference in the discussion…the patients also had alcoholic cardiomyopathy and what is the LV Ejection Fraction of these patients compared to those who did not have alcoholic cirrhosis?
Response: NIS database doesnt record echo parameters like LVEF. Alcoholic cardiomyopathy did not have its own specific ICD 10 code hence was not able to be addressed in the mortality proporitons but this was accounted for in the multivariage logistic regression model ( chronic alcohol abuse) to account for its confounding effect
In Tables 3 and 4, why is the column referring to the value of p in the 3rd column and not in the 4th column? Next to the 95% confidence interval column?
Response: table has been redesigned
For all other recommendations for Discussion section and results, will make changes accordingly
Line?: “and diabetes. (Exact matching was done for Age, race, smoking, alcohol abuse and CKD). Probit…”; must be exact…age.
Response: change has been made
propensity-matched and multivariate logistic regression analysis of 7595 patients with both ADHF and cirrhosis showed that mortality was almost two times higher in patients with ADHF and cirrhosis compared to patients without cirrhosis (p<0.001), mean hospital length of stay was 1.2 times higher in patients with ADHF and cirrhosis (p\0.001), but the mean total burden of hospitalization was not statistically significant.”
Response: changed
but what are these pathological differences…myocardial depression and structural changes induced by ethanol in the myocardium? Discuss this aspect better and better explain the potential pathophysiological differences…what mechanisms?
Response: explained this statement quoting alcoholic dilated cardiomyopahty, and general raised mortality due to alcohol abuse
Lines?: it reads “As mentioned in the introduction section, CLD playing a role in medication non-adherence, especially ACE/ARBs in the management of systolic heart failure, there has been several studies looking into pathophysiologic mechanisms of cirrhosis as well as Non-Alcoholic Fatty Liver Diseases (NAFLD) having negative impact on the left ventricular function, adding further molecular level evidence for the findings we have in our study.”; this sentence is very confusing and We don't understand the interconnection that the authors want to make...it must be completely modified and give it a logical sequence; only to ACEi/ARB???; what molecular level evidence? (describe related molecular changes…).
Response: this statement has been omitted. Non adherence with ACE/ ARB in CLD was noted to be due to hypotension preventing its initiation and continuation ( as RAAS systemt is responsible for maintaining systolic blood pressure in the generalzied vasodilatory state due to CLD
Lines?: reads “. In the literature review by Mantovni et al. [14], it was noted that adult individuals with NAFLD in the absence of severe obesity, hypertension, and diabetes have mildly increased left ventricular mass and early features of diastolic dysfunction”; but in this article by Mantovani et al (2016) you can read Spectrum of the most important cardiac complications associated with NAFLD: Coronary heart disease, cardiac hypertrophy, heart failure, atrial fibrillation, aortic valve sclerosis, mitral annulus calcification, QTc prolongation”, ,,; Therefore, the text must be completed with the spectrum of diseases associated with NAFLD, correcting to Mantovani (instead of Mantovni); It is also not understood what the relationship between NAFLD is and the patients included in this study as in the methods chapter there is no reference to the number of patients included in the study with the aforementioned pathology (NAFLD.)
Response: change has been made
Lines?: it reads “The pathological effects of Liver cirrhosis on cardiac musculature were analyzed further in a real-world study by VanWagner et al. [16], it was noted that patients with NAFLD had lower early diastolic relaxation(e’) velocity (10.8 +/- 2.6 Vs 11.9 +/…”; but cirrhosis or NAFLD…
Response: sub stratification based on NAFLD was not made
- Lines: it reads “More affluent neighborhoods are associated with better health outcomes, while disadvantaged areas often have worse outcomes11???”. What does this have to do with the current study? (no results are mentioned in this regard)…
- Response: changes made
- Lines: It reads “especially the controversial syndrome X. Kim et al. [21]. “Syndrome X??? What syndrome is this coronary syndrome X????
- Response: changed to metabolic syndrome X
Reviewer 2 Report
Comments and Suggestions for Authors
Dear Authors, I was reviewing with interest the mansucript entitled "Outcomes in Congestive Heart Failure Admissions with Chronic Liver Disease: A Nationwide Analysis Using the National In-Patient Sample". You deal with an interesting subject on the basis of a huge database. The findings add to our knowledge about patients with ADHF in combination with CLD. The methods used to rule out confounding factors and bias seem to be appropriate. However there remain some minor shortcomings: You miss completely the hepato-renal syndrome which has great influence on the outcome of HF (line 81), especially in the discussion. Additionally the abbreviation "NIS" in line 12 should be explained. The literature section is adequate and well balanced.
Author Response
We avoided patients with HRS during the admission as the database was subject to confounding regarding all diagnoses other than the main admission diagnosis, as temporal relation could not be stablished. Ex: if HRS is inlcuded using the ICD 10 code, the database did not let us find whether it was a diagnosis related to the current admission or a past diagnosis
Round 2
Reviewer 1 Report
Comments and Suggestions for Authors
Yes, I accept.